

# Molecular identification and expression patterns of odorant binding protein and chemosensory protein genes in *Athetis lepigone* (Lepidoptera: Noctuidae)

Ya-Nan Zhang[1,2], Xiu-Yun Zhu[1], Ji-Fang Ma[3], Zhi-Ping Dong[3], Ji-Wei Xu[1], Ke Kang[4] and Long-Wa Zhang[5]

[1] College of Life Sciences, Huaibei Normal University, Huaibei, China

[2] Key Laboratory of Integrated Pest Management on Crops in East China, Ministry of Agriculture, Key Laboratory of Integrated Management of Crop Diseases and Pests, Ministry of Education, College of Plant Protection, Nanjing Agricultural University, Nanjing, China

[3] Institute of Millet Crops, Hebei Academy of Agriculture and Forestry Sciences, Shijiazhuang, China

[4] Anhui Vocational & Technical College of Forestry, Hefei, China

[5] Anhui Provincial Key Laboratory of Microbial Control, School of Forestry & Landscape Architecture, Anhui Agricultural University, Hefei, China

## ABSTRACT

The olfaction system of insects plays an important role in mediating various physiological behaviors, including locating hosts, avoiding predators, and recognizing mates and oviposition sites. Therefore, some key genes in the system present valuable opportunities as targets for developing novel green pesticides. *Athetis lepigone*, a noctuid moth can feed on more than 30 different host plants making it a serious polyphagous pest worldwide, and it has become one of the major maize pests in northern China since 2011. However, there are no reports on effective and environmentally friendly pesticides for the control of this pest. In this study, we identified 28 genes encoding putative odorant binding proteins (OBPs) and 20 chemosensory protein (CSPs) genes based on our previous *A. lepigone* transcriptomic data. A tissue expression investigation and phylogenetic analysis were conducted in an effort to postulate the functions of these genes. Our results show that nearly half (46.4%) of the *AlOBPs* exhibited antennae-biased expression while many of the *AlCSPs* were highly abundant in non-antennal tissues. These results will aid in exploring the chemosensory mechanisms of *A. lepigone* and developing environmentally friendly pesticides against this pest in the future.

## INTRODUCTION

The olfaction system of insects mediates a host of physiological behaviors, such as host location, predator avoidance, and mate and oviposition site recognition (*Leal, 2013*). Many studies show that the periphery process of insect olfaction requires a set of genes, including those that encode odorant binding proteins (OBPs), chemosensory proteins (CSPs), and chemosensory receptors (*Elfekih et al., 2016*; *Glaser et al., 2015*; *Larter, Sun &*

Corresponding authors
Ya-Nan Zhang,
ynzhang_insect@163.com
Long-Wa Zhang,
zhanglw@ahau.edu.cn,
longwazhang@126.com

*Carlson, 2016*; *Li et al., 2015*; *Paula et al., 2016*; *Zhang et al., 2013*). Generally, OBPs/CSPs located in the antennal sensillar lymph can recognize and bind external odorants that can then be transferred by OBPs/CSPs through the sensillar lymph to chemosensory receptors, odorant receptors (ORs) and ionotropic receptors (IRs). Therefore, OBPs and CSPs play key roles in helping insects recognize various odorants and regulate their behaviors (*Dani et al., 2011*; *Zhou, 2010*). These functions also suggest that these protein families may present valuable opportunities as target genes for developing novel green pesticides.

Insect OBPs are a class of small, abundant, and water-soluble extracellular proteins of ~14 KDa. Most OBPs use six positionally conserved cysteines to form three interlocking disulphide bridges that stabilize the protein's three-dimensional structure (*Lagarde et al., 2011*; *Leal, Nikonova & Peng, 1999*; *Pelosi & Maida, 1995*; *Vogt & Riddiford, 1981*). Since the first OBP was identified in *Antheraea polyphemus* (*Vogt & Riddiford, 1981*), many OBPs have been found in various insects based on genomic or transcriptomic methods in recent years. Based on the structural features and similarity in protein sequences, insect OBPs can be divided into three major subclasses (*Li et al., 2013*; *Schultze et al., 2012*; *Spinelli et al., 2012*; *Zhou, 2010*): classic OBPs, including pheromone binding proteins (PBPs), general odorant binding proteins (GOBPs), and two OBPs involved in the recognition of female sex pheromones and host volatiles; plus-C OBPs; and minus-C OBPs, which may also participate in the binding of host volatiles as suggested by an *in vitro* competitive binding assay.

Olfactory specific protein D (OS-D), the first insect CSP gene, was discovered in *Drosophila melanogaster* (*McKenna et al., 1994*). By using similar methods as for OBP identification, many CSPs have been discovered in distinct insects (*Guo et al., 2011*; *Iovinella et al., 2013*; *Jacquin-Joly et al., 2001*; *Liu et al., 2010*; *Missbach et al., 2015*; *Picimbon et al., 2001*; *Wanner et al., 2004*). Unlike OBPs, CSPs are smaller and more conserved in distinct insects, which only have four conserved cysteines that form two interlocking disulphide bridges (*Bohbot et al., 1998*; *Lartigue et al., 2002*; *Maleszka & Stange, 1997*; *Pelosi, Calvello & Ban, 2005*; *Zhang et al., 2014*). Furthermore, OBPs are usually specifically or predominately expressed in the antennae, whereas many CSPs are expressed in the antennae and other tissues (*Pelosi, Calvello & Ban, 2005*; *Vogt, 2005*; *Zhang et al., 2016a*; *Zhang et al., 2013*), suggesting insect CSPs have both chemosensation and non-chemosensation functions as is illustrated by their association with chemosensation in moths (*Jacquin-Joly et al., 2001*; *Sun et al., 2015*; *Zhang et al., 2014*), limb regeneration in *Periplaneta eparate* (*Nomura et al., 1992*), embryo development in *Apis mellifera* (*Maleszka et al., 2007*), behavioral phase change in *Locusta migratoria* (*Guo et al., 2011*), and female moth survival and reproduction in *Spodoptera exigua* (*Gong et al., 2012*).

*Athetis lepigone* Möschler (Lepidoptera: Noctuidae) is a serious polyphagous pest found worldwide (*Fu et al., 2014*; *Karsholt, Van Nieukerken & De Jong, 2013*; *Lindeborg, 2008*; *Nikolaevitch & Vjatcheslavovna, 2003*; *Zhang, Zhao & Ding, 2009*) that can feed on more than 30 different host plants species and has become one of the major maize pests in northern China since 2011 (*Jiang et al., 2011*; *Ma et al., 2012*; *Shi et al., 2011*). However, there are no reports on the chemosensory mechanism mediated by OBPs/CSPs between the pests and host plants. In this study, we identified 28 and 20 genes encoding putative

AlOBPs (*A. lepigone* OBPs) and AlCSPs (*A. lepigone* CSPs), respectively, based on our previous transcriptomic data of *A. lepigone* (*Zhang et al., 2016b*). Tissue expression and phylogenetic analyses were conducted in an effort to postulate the function of these genes. We found that most AlOBPs and AlCSPs had high identities with those in other moths (*Campanacci et al., 2001*; *Gu et al., 2013*; *Liu et al., 2015c*; *Zhang et al., 2015a*; *Zhang et al., 2013*; *Zhang et al., 2015b*); nearly half of the *AlOBPs* exhibited antennae-biased expression, and many *AlCSP*s were found in various tissues and were highly expressed in proboscises, legs, and wings, which will help us explore the chemosensory mechanism of *A. lepigone* and develop environmentally friendly pesticides against this pest in the future.

## MATERIALS & METHODS

### Insect rearing and tissue collection

*A. lepigone* were fed an noctuid artificial diet (*Huang et al., 2002*) at a temperature of 26 $\pm$ 1 °C in a 14:10 h, light:dark photoperiod. Pupae were sexed, and males and females were placed into separate enclosures. Adult moths were given a 10% honey solution after emergence. We collected 25–30 female antennae (FA), 25–30 male antennae (MA), 50–60 proboscises (Pr, ♀:♂ = 1:1), 10–12 abdomen (Ab, ♀:♂ = 1:1), 28–30 legs (Le, ♀:♂ = 1:1), and 28–30 wings (Wi, ♀:♂ = 1:1) from three-day-old virgin adults. All samples were immediately frozen in liquid nitrogen and stored at −80 °C until use.

### RNA isolation and cDNA synthesis

Total RNA was extracted using the MiniBEST Universal RNA Extraction Kit (TaKaRa, Dalian, China) following the manufacturer's instructions, and the RNA quality was checked using a spectrophotometer (NanoDrop$^{TM}$ 2000; Thermo Fisher Scientific, USA). The single-stranded cDNA templates were synthesized from 1 μg total RNA from various tissue samples using the PrimeScript$^{TM}$ RT Master Mix (TaKaRa, Dalian, China).

### Sequence analyses

The open reading frames (ORFs) of the putative chemosensory genes were predicted using ORF Finder (http://www.ncbi.nlm.nih.gov/gorf/gorf.html). The similarity searches were performed with NCBI-BLAST (http://blast.ncbi.nlm.nih.gov/). Putative N-terminal signal peptides for AlOBPs and AlCSPs were predicted by SignalP 4.1 (http://www.cbs.dtu.dk/services/SignalP/) (*Petersen et al., 2011*).

### Phylogenetic analysis

Phylogenetic trees were reconstructed for the analysis of *AlOBPs* and *AlCSPs*, based on the gene sequences of *A. lepigone* and those of other insects. The OBP data set contained 28 sequences from *A. lepigone* (Table S1), and 100 from other insects including *Bombyx mori* (*Gong et al., 2009*), *Manduca sexta* (*Grosse-Wilde et al., 2011*), *Sesamia inferens* (*Zhang et al., 2013*), and *Spodoptera littoralis* (*Legeai et al., 2011*). The CSP dataset contained 20 sequences from *A. lepigone* (Table S1) and 68 from other insects including *B. mori* (*Gong et al., 2007*), *M. sexta* (*Grosse-Wilde et al., 2011*), *S. inferens* (*Zhang et al., 2013*), and *S. littoralis* (*Legeai et al., 2011*). Amino acid sequences were aligned with MAFFT version

7 (http://mafft.cbrc.jp/alignment/server/), and phylogenetic trees were constructed using PhyML (*Guindon et al., 2010*) based on the LG substitution model (*Le & Gascuel, 2008*) with Nearest Neighbour Interchange (NNI), and branch support estimated by a Bayesian-like transformation of the aLRT (aBayes) method. Dendrograms were created and colored in FigTree (http://tree.bio.ed.ac.uk/software/figtree/).

## Quantitative real time-PCR

Expression profiling of *AlOBPs* and *AlCSPs* was performed using quantitative real time-PCR (qRT-PCR) performed in a LightCycler® 96 (Roche, Switzerland) with a mixture of 5 μL 2X SYBR® Premix Ex Taq (Tli RNaseH Plus) (TaKaRa, Dalian), 0.2 μL of each primer (10 μM), 2.5 ng of sample cDNA, and 3.6 μL of sterilized ultrapure $H_2O$. The reaction program was as follows: 30 s at 95 °C, 40 cycles of 95 °C for 5 s, and 60 °C for 20 s. The results were analyzed using a LightCycler® 96 SW 1.1. The qRT-PCR primers (Table S2) were designed with Beacon Designer 7.9 (PREMIER Biosoft International, CA, USA). This was followed by the measurement of fluorescence over a 55 to 95 °C melting curve to detect a single gene-specific peak and to check the absence of primer dimer peaks, and a single and discrete peak was detected for all primers tested. Negative controls consisted of non-template reactions where the cDNA was replaced with $H_2O$.

Expression levels of *AlOBPs* and *AlCSPs* were calculated relative to the reference genes *AlGAPDH* (*A. lepigone* glyceraldehyde-3-phosphate dehydrogenase) and *AlEF* (*A. lepigone* elongation factor-1 alpha) using the Q-Gene method in the Microsoft Excel-based software Visual Basic (*Muller et al., 2002*; *Simon, 2003*). For each sample, three biological replicates were performed with three technical replicates per biological replicate.

## Statistical analysis

Data (mean ± SE) from various samples were subjected to one-way nested analysis of variance (ANOVA) followed by a least significant difference test (LSD) for mean comparisons using the SPSS Statistics 22.0 software (SPSS Inc., Chicago, IL, USA).

## RESULTS

### Identification of putative OBP genes in *A. lepigone*

Based on our previous antennal transcriptomic data (NCBI-SRX number: 2543665) for *A. lepigone* (*Zhang et al., 2016b*), we first identified 28 genes encoding putative OBPs including three *PBP*s and two *GOBP*s (Table 1). Among the 28 *AlOBP*s, 24 sequences were predicted to be full-length genes that encoded 133 to 246 amino acids; all 24 genes had a predicted signal peptide at the N-terminus. According to the number and position of conserved cysteines, insect OBPs can be divided into different subclasses: classic OBPs, Plus-C OBPs, and Minus-C OBPs (*Zhou, 2010*). Here, AlOBP4 and AlOBP9 had no conserved cysteines at the C2 and C5 positions, and, therefore, belonged to the Minus-C OBP subfamily; AlOBP2, AlOBP7, and AlOBP14 had cysteines in addition to the six conserved cysteines; therefore, they belonged to the Plus-C OBP subfamily; the other 19 full-length AlOBPs belonged to the Classic OBP subfamily, which had six conserved cysteines at the corresponding positions (Fig. S1).

Zhang et al. (2017), *PeerJ*, DOI 10.7717/peerj.3157

**Table 1** The BLASTX match of OBP genes in *A. lepigone*.

| Gene name | ORF (aa) | Signal peptide | Complete ORF | Best blastx match | | | | |
|---|---|---|---|---|---|---|---|---|
| | | | | Name | Acc. No. | Species | E value | Identity (%) |
| GOBP1 | 163 | 1-18 | Y | general odorant binding protein 1 | ABI24160.1 | *Agrotis ipsilon* | 8.00E–83 | 95 |
| GOBP2 | 162 | 1-21 | Y | general odorant binding protein 2 | AHC72380.1 | *Sesamia inferens* | 2.00E–92 | 91 |
| PBP1 | 167 | 1-23 | Y | pheromone binding protein 1 precursor | AAC05702.2 | *Mamestra brassicae* | 3.00E–88 | 90 |
| PBP2 | 170 | 1-24 | Y | pheromone binding protein 2 precursor | AAC05701.2 | *Mamestra brassicae* | 5.00E–58 | 90 |
| PBP3 | 164 | 1-22 | Y | pheromone-binding protein 3 | AFM36758.1 | *Agrotis ipsilon* | 2.00E–85 | 90 |
| OBP1 | 116 | N | N | SexiOBP14 | AGP03460.1 | *Spodoptera exigua* | 7.00E–54 | 88 |
| OBP2 | 146 | 1-17 | Y | odorant binding protein 6 | AGR39569.1 | *Agrotis ipsilon* | 2.00E–84 | 88 |
| OBP3 | 120 | N | N | odorant binding protein 8 | AKI87969.1 | *Spodoptera litura* | 5.00E–79 | 85 |
| OBP4 | 138 | 1-16 | Y | odorant-binding protein 18 | AFI57167.1 | *Helicoverpa armigera* | 2.00E–52 | 85 |
| OBP5 | 147 | 1-21 | Y | pheromone binding protein 4 | AAL66739.1 | *Mamestra brassicae* | 1.00E–81 | 84 |
| OBP6 | 134 | 1-17 | Y | ABPX | AGS36754.1 | *Sesamia inferens* | 2.00E–54 | 83 |
| OBP7 | 203 | 1-20 | Y | odorant-binding protein 19 | AGC92793.1 | *Helicoverpa assulta* | 2.00E–69 | 83 |
| OBP8 | 147 | 1-20 | Y | oderant binding protein 6 | AFM77984.1 | *Spodoptera exigua* | 4.00E–56 | 82 |
| OBP9 | 133 | 1-16 | Y | odorant binding protein 9 | AGH70105.1 | *Spodoptera exigua* | 5.00E–84 | 80 |
| OBP10 | 96 | N | N | odorant binding protein 1 | AGR39564.1 | *Agrotis ipsilon* | 2.00E–58 | 79 |
| OBP5 | 147 | 1-21 | Y | pheromone binding protein 4 | AAL66739.1 | *Mamestra brassicae* | 1.00E–81 | 84 |
| OBP11 | 152 | 1-21 | Y | pheromone binding protein 4 | AAL66739.1 | *Mamestra brassicae* | 1.00E–30 | 78 |
| OBP12 | 141 | 1-26 | Y | odorant binding protein 8 | AGH70104.1 | *Spodoptera exigua* | 9.00E–78 | 77 |
| OBP13 | 184 | 1-20 | Y | odorant binding protein | AII00978.1 | *Dendrolimus houi* | 1.00E–106 | 75 |
| OBP14 | 186 | 1-17 | Y | odorant binding protein 1 | AGR39564.1 | *Agrotis ipsilon* | 8.00E–97 | 75 |
| OBP15 | 155 | 1-24 | Y | SexiOBP11 | AGP03457.1 | *Spodoptera exigua* | 2.00E–82 | 73 |
| OBP16 | 148 | 1-21 | Y | OBP7 | AEB54591.1 | *Helicoverpa armigera* | 7.00E–54 | 70 |
| OBP17 | 246 | 1-19 | Y | odorant binding protein | AII00994.1 | *Dendrolimus kikuchii* | 2.00E–74 | 67 |
| OBP18 | 149 | 1-22 | Y | OBP5 | AEB54581.1 | *Helicoverpa armigera* | 8.00E–58 | 65 |
| OBP19 | 71 | 1-22 | N | OBP6 | AGS36748.1 | *Sesamia inferens* | 2.00E–25 | 65 |
| OBP20 | 170 | 1-23 | Y | odorant binding protein 4 | AKI87965.1 | *Spodoptera litura* | 2.00E–76 | 61 |
| OBP21 | 153 | 1-21 | Y | SexiOBP9 | AGP03455.1 | *Spodoptera exigua* | 2.00E–77 | 59 |
| OBP22 | 146 | 1-25 | Y | SexiOBP12 | AGP03458.1 | *Spodoptera exigua* | 1.00E–72 | 58 |
| OBP23 | 145 | 1-17 | Y | odorant binding protein | ADY17886.1 | *Spodoptera exigua* | 1.00E–85 | 40 |

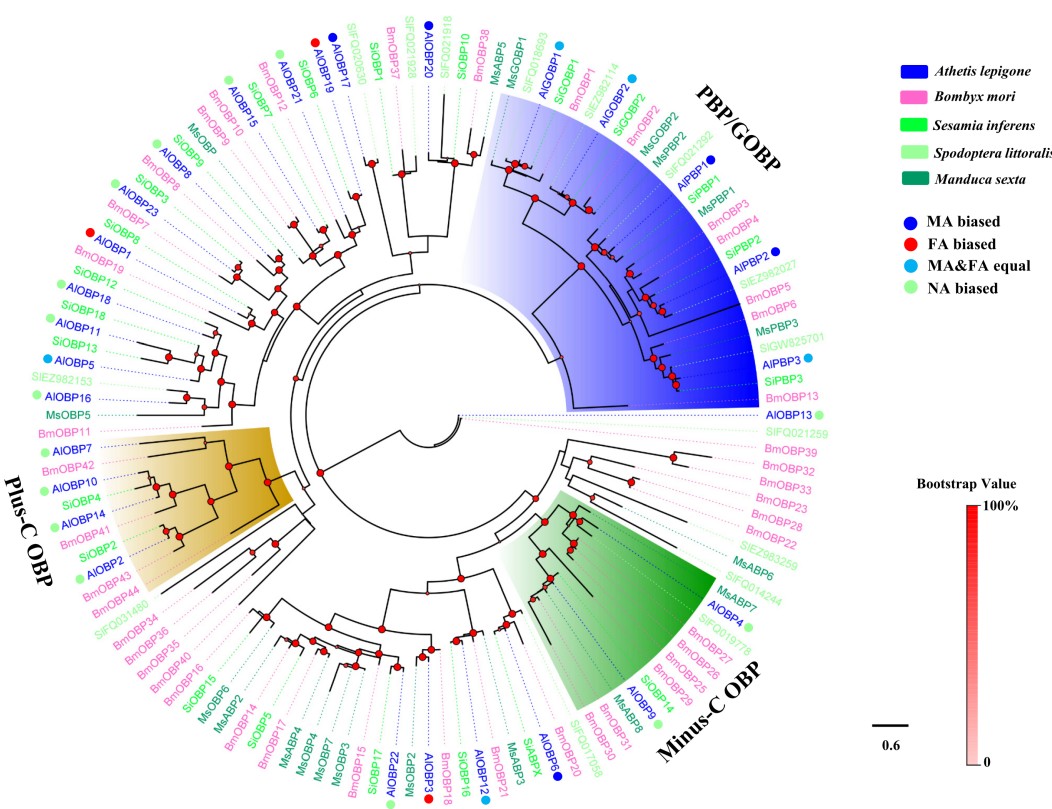

**Figure 1** **Phylogenetic tree of moth OBPs.** The *A. lepigone* translated genes are shown in blue. This tree was constructed using phyML based on alignment results of MAFFT. Al, *A. lepigone*; Bm, *B. mori*; Si, *S. inferens*; Sl, *S. littorali*; Ms, *M. sexta*.

## Identification of putative CSP genes in *A. lepigone*

Twenty putative genes encoding CSPs were identified in *A. lepigone* via antennal transcriptome analysis (Table 2). Eighteen of these had full length ORFs with 4 conserved cysteines in corresponding positions (Fig. S2), and seventeen genes (except *AlCSP14*) had a predicted signal peptide at the N-terminus. The results of a BLASTX match showed that 80% of these CSPs ($n = 16$) had >70% identity with other CSPs from different moths and that this was higher than the sequence identities of the OBPs (75%) (Table 2).

## Phylogenetic analyses of moth OBPs and CSPs

Two phylogenetic trees, one of moth OBPs and one of moth CSPs, were constructed using protein sequences from *A. lepigone*, *B. mori*, *S. inferens*, *S. littoralis*, and *M. sexta*, (Figs. 1 and 2). Similar to other studies (*He et al., 2010*; *Pelosi et al., 2014b*; *Vogt, Rybczynski & Lerner, 1991*; *Xiu & Dong, 2007*), the OBP tree showed that moth OBPs can be divided into PBP/GOBP, Minus-C OBP, and Plus-C OBP subfamilies. AlPBP1-3 clustered into the PBP subfamily and the AlGOBPs (1 and 2) clustered into the GOBP subfamily. Two AlOBPs (AlOBP4 and AlOBP9) clustered into the moth Minus-C OBP subfamily, and four AlOBPs (AlOBP2, AlOBP7, AlOBP10, and AlOBP14) clustered into the moth Plus-C OBP subfamily. The rest of the AlOBPs clustered with at least one orthologous moth gene. In

Zhang et al. (2017), *PeerJ*, DOI 10.7717/peerj.3157

**Table 2  The BLASTX match of CSP genes in *A. lepigone*.**

| Gene name | ORF (aa) | Signal peptide | Complete ORF | Best blastx match | | | | |
|---|---|---|---|---|---|---|---|---|
| | | | | Name | Acc. No. | Species | E value | Identity (%) |
| CSP1 | 124 | 1-16 | Y | chemosensory protein 15 | AGH20053.1 | *Helicoverpa armigera* | 5.00E–62 | 83 |
| CSP2 | 124 | 1-15 | Y | chemosensory protein precursor | NP_001037066.1 | *Bombyx mori* | 2.00E–38 | 61 |
| CSP3 | 122 | 1-18 | Y | ejaculatory bulb-specific protein 3-like | XP_012549936.1 | *Bombyx mori* | 2.00E–45 | 74 |
| CSP4 | 294 | 1-16 | Y | chemosensory protein | AIW65104.1 | *Helicoverpa armigera* | 2.00E–130 | 78 |
| CSP5 | 56 | N | N | chemosensory protein | AII01011.1 | *Dendrolimus houi* | 3.00E–17 | 62 |
| CSP6 | 150 | 1-19 | Y | putative chemosensory protein | AGY49270.1 | *Sesamia inferens* | 6.00E–72 | 78 |
| CSP7 | 114 | 1-19 | Y | sensory appendage protein-like protein | AAK14793.1 | *Mamestra brassicae* | 1.00E–28 | 61 |
| CSP8 | 127 | 1-18 | Y | chemosensory protein 6 | AGR39576.1 | *Agrotis ipsilon* | 5.00E–63 | 91 |
| CSP9 | 127 | 1-16 | Y | chemosensory protein | AAF71289.1 | *Mamestra brassicae* | 3.00E–59 | 83 |
| CSP10 | 123 | 1-18 | Y | chemosensory protein 8 | AGR39578.1 | *Agrotis ipsilon* | 4.00E–68 | 85 |
| CSP11 | 123 | 1-16 | Y | chemosensory protein | AIW65100.1 | *Helicoverpa armigera* | 2.00E–65 | 76 |
| CSP12 | 128 | 1-18 | Y | chemosensory protein CSP2 | ABM67689.1 | *Spodoptera exigua* | 4.00E–70 | 81 |
| CSP13 | 123 | 1-19 | Y | chemosensory protein | AIX97829.1 | *Cnaphalocrocis medinalis* | 1.00E–56 | 81 |
| CSP14 | 46 | N | N | putative chemosensory protein | AGY49260.1 | *Sesamia inferens* | 3.00E–25 | 100 |
| CSP15 | 122 | 1-16 | Y | chemosensory protein 10 | AFR92094.1 | *Helicoverpa armigera* | 1.00E–73 | 89 |
| CSP16 | 130 | N | Y | chemosensory protein 15 | NP_001091781.1 | *Bombyx mori* | 3.00E–42 | 59 |
| CSP17 | 127 | 1-18 | Y | putative chemosensory protein | AGY49267.1 | *Sesamia inferens* | 1.00E–70 | 81 |
| CSP18 | 123 | 1-18 | Y | chemosensory protein 8 | AFR92092.1 | *Helicoverpa armigera* | 8.00E–43 | 74 |
| CSP19 | 120 | 1-16 | Y | chemosensory protein 4 | AGR39574.1 | *Agrotis ipsilon* | 1.00E–60 | 79 |
| CSP20 | 107 | 1-18 | Y | chemosensory protein 5 | AGR39575.1 | *Agrotis ipsilon* | 4.00E–53 | 97 |
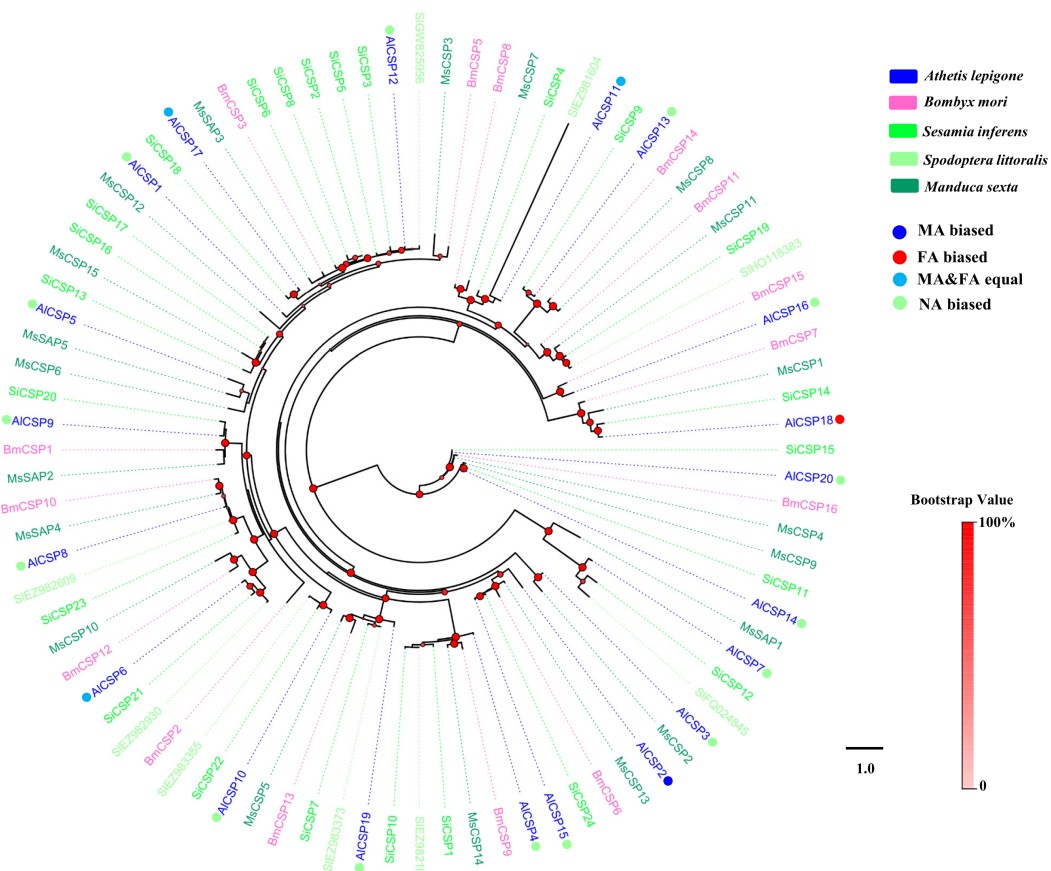

**Figure 2 Phylogenetic tree of moth CSPs.** The *A. lepigone* translated genes are shown in blue. This tree was constructed using phyML based on alignment results of MAFFT. Al, *A. lepigone*; Bm, *B. mori*; Si, *S. inferens*; Sl, *S. littorali*; Ms, *M. sexta*.

the constructed CSP tree, our results indicated that all 20 AlCSPs were distributed along various branches and each clustered with at least one other moth ortholog.

## OBPs and CSPs expression patterns in *A. lepigone*

We used the qRT-PCR results to investigate the expression profiles of all *AlOBP*s and *AlCSP*s. The results showed that all the OBPs and CSPs were expressed in the adult antennae of *A. lepigone*. Among the 28 *AlOBP*s, 13 *AlOBP*s (including PBPs and GOBPs) were significantly highly expressed in the antennae ($p < 0.05$, ANOVA, LSD), including 5 male-biased (*AlPBP1, AlPBP2, AlOBP6, AlOBP17,* and *AlOBP20*) and 3 female-biased (*AlOBP1, AlOBP3,* and *AlOBP19*) OBP genes. In all 28 *AlOBP*s, *AlGOBP1* and *AlPBP1* (male antennae) exhibited the highest expression levels, and *AlOBP19* exhibited the lowest expression abundance (Fig. 3). In addition, eight *AlOBP*s (*AlOBP4, 8, 11, 14, 16, 21, 22,* and *23*) exhibited proboscis-biased expression, *AlOBP10* was expressed significantly more in the adult abdomen, and four *AlOBP*s (*AlOBP2, 9, 13,* and *18*) displayed higher expression levels in adult wings than in other tissues (Fig. 3).

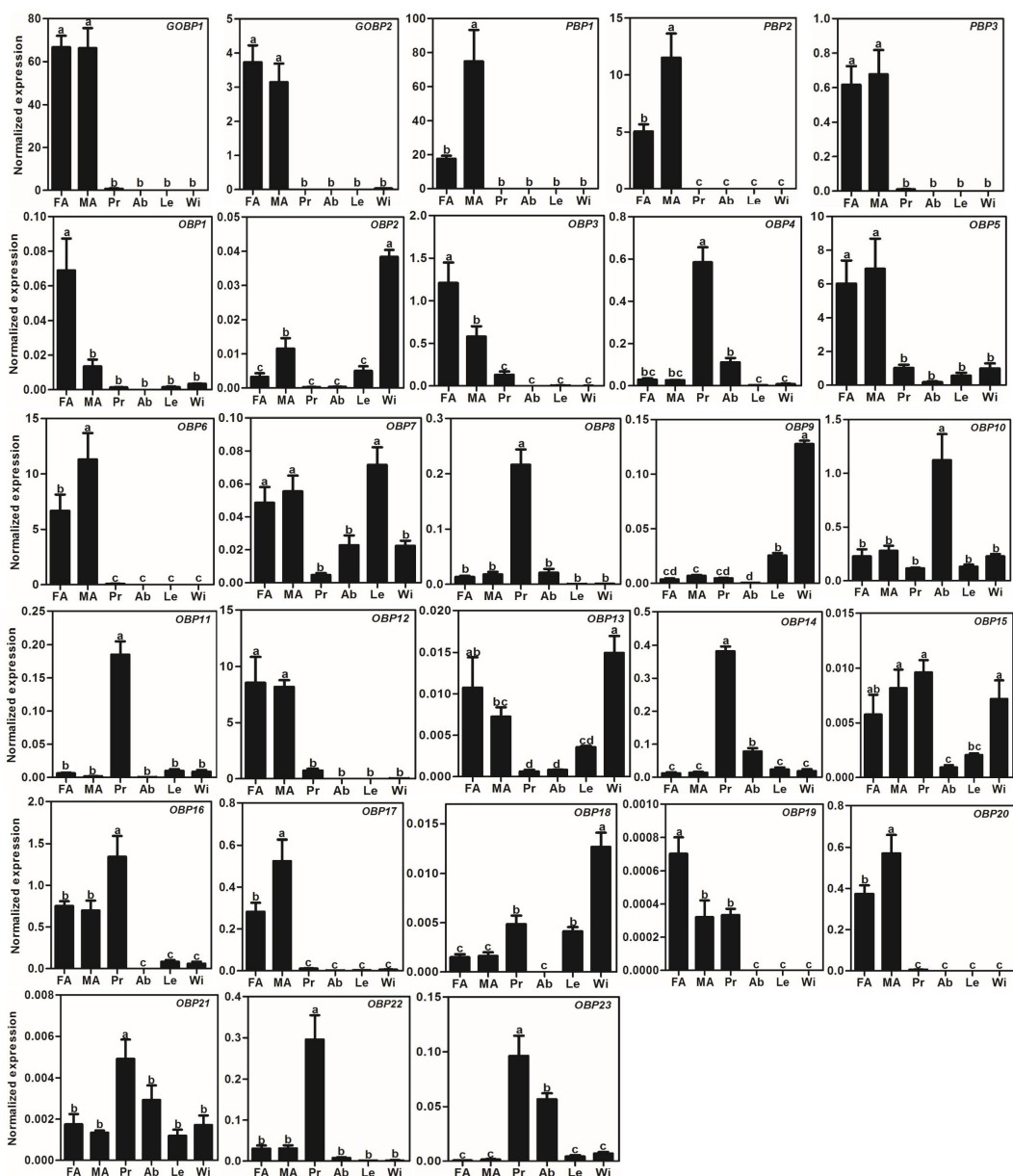

**Figure 3** **Expression patterns of OBP genes in *A. lepigone*.** The relative expression level is indicated as mean ± SE (*N* = 3). Different capital letters mean significant difference between tissues (*p* < 0.05, ANOVA, LSD). FA, female antennae; MA, male antennae; Pr, proboscises; Ab, abdomen; Le, legs; Wi, wings.

Compared to *AlOBPs*, *AlCSPs* were highly expressed in adult antennae as well as in non-antennae tissues. Of the 20 identified *AlCSP* genes, only *AlCSP2, AlCSP6,* and *AlCSP18* had antennae-biased expression; *AlCSP2* was male-biased and *AlCSP18* was female-biased in their expression. Six *AlCSP* genes (*AlCSP1, 9, 12, 15, 16* and *20*) were highly expressed in the proboscises, and nine (*AlCSP3-5, 7, 8, 10, 13, 14* and *19*) were highly expressed in the wings; among the 20 total *AlCSPs*, *AlCSP14* and *AlCSP5* displayed the highest and lowest expression levels in the antennae, respectively (Fig. 4).

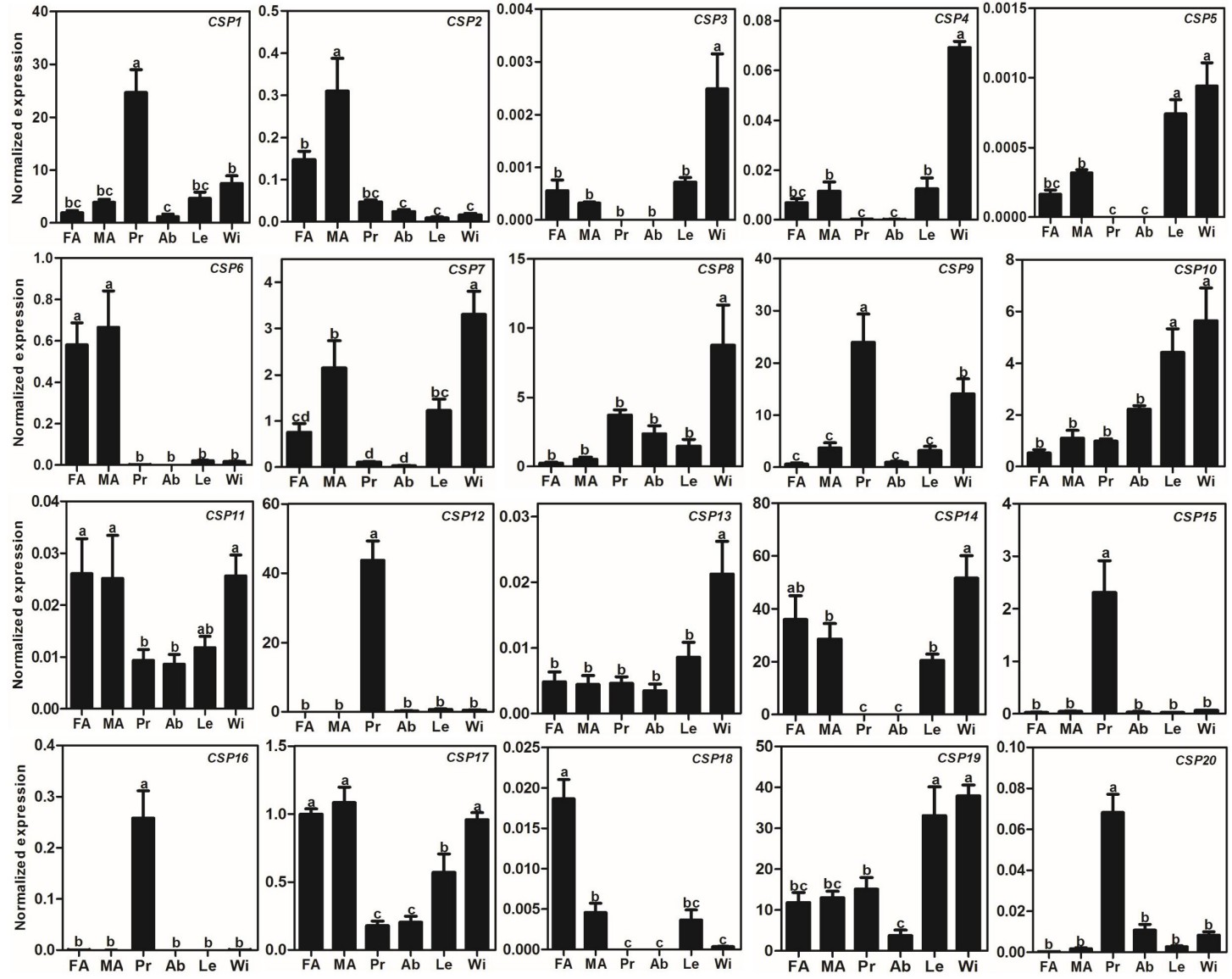

**Figure 4** **Expression patterns of CSP genes in *A. lepigone*.** The relative expression level is indicated as mean $\pm$ SE ($N = 3$). Different capital letters mean significant difference between tissues ($p < 0.05$, ANOVA, LSD). FA, female antennae; MA, male antennae; Pr, proboscises; Ab, abdomen; Le, legs; Wi, wings.

## DISCUSSION

In this study, we first identified 28 and 20 genes encoding putative AlOBPs and AlCSPs, respectively, based on our previous *A. lepigone* transcriptomic data (*Zhang et al., 2016b*). The number of *AlOBP* and AlCSP genes identified for this species are similar to some reported moths, such as *C. suppressalis* (*Cao et al., 2014*), *H. armigera* (*Liu et al., 2012*), and *B. mori* (*Gong et al., 2007*), but there are certain different from *S. litura* (*Gu et al., 2015*), *S. inferens* (*Zhang et al., 2013*), *H. armigera* (*Liu et al., 2012*) and *B. mori* (*Gong et al., 2009*). The reasons for the differences in gene number may be due to: (1) the different

chemosensory behaviors of different moths requiring distinct molecular mechanisms that have developed over evolutionary time; (2) the genomic data will help us identify more genes from *A. lepigone* as well as from other moths in the future.

Many studies have shown that insect OBPs are mainly expressed in the antennae of both sexes and that they may play key roles in the process of host location, mating, and oviposition by allowing the insect to accurately recognize environmental odorants (*Larter, Sun & Carlson, 2016*; *Leal, 2013*; *Qiao et al., 2009*; *Zhou et al., 2009*). The phylogenetic tree of moth OBPs showed that AlOBPs were divided into different subfamilies, including the PBP/GOBP, Minus-C OBP, and Plus-C OBP proteins suggesting that the structural diversity of AlOBPs may be involved in chemosensation and/or in other physiological processes. Based on the qRT-PCR analyses, we found that 46% of the 28 *AlOBP*s were highly expressed in the antennae indicating that these AlOBP proteins have putative roles in the odorant reorganization of *A. lepigone*. Similar to our previous work and to other studies (*Gu et al., 2015*; *McKenzie, Oxley & Kronauer, 2014*; *Zhang et al., 2016a*; *Zhang et al., 2013*), we found that there were five *AlOBP* genes highly expressed in non-antennal tissues (legs and wings), including one abdomen-biased AlOBP-encoding gene and four wing-biased *AlOBP* genes, indicating that these OBPs may have other non-chemosensory functions.

Five AlPBP/GOBPs displayed higher expression in the adult antenna (especially *AlGOBP1* and *AlPBP1*), which is consistent with that reported for PBP/GOBPs in other moths (*Liu, Liu & Dong, 2013*; *Liu et al., 2015b*; *Zhang et al., 2013*). According to recent functional studies of moth PBP/GOBPs (*Jin et al., 2014*; *Liu et al., 2015a*; *Liu, Liu & Dong, 2013*; *Sun, Liu & Wang, 2013*; *Zhu et al., 2016*) and *D. melanogaster* LUSH protein (OBP76a) (*Ha & Smith, 2006*; *Stowers & Logan, 2008*; *Zhou et al., 2004*), we hypothesize that the AlPBP/GOBPs may also play important roles in recognizing the sex pheromones of female moths and some host plant volatiles. Additionally, there are three male-biased and three female-biased AlOBP genes, indicating that these sex-biased OBPs may participate in the reorganization of female or male sex pheromones, plant volatiles from oviposition sites, or other sex-related odorants, and need further analysis to explore their exact roles such as through fluorescence competitive binding assays (*Liu et al., 2015b*), CRISPR/Cas9 mediated genome editing (*Zhu et al., 2016*), and gene mutations (*Stowers & Logan, 2008*).

Studies on *CSP* genes in certain insects have shown that they are smaller and more conserved than *OBP* genes and that they are widely expressed in different parts of the insect body (*Calvello et al., 2005*; *Gong et al., 2007*; *Pelosi et al., 2014a*; *Zhang et al., 2013*). Our BLASTX results showed that the AlCSPs had relatively high identities with other moth CSPs indicating high conservation of CSPs among moths. Our results agreed with those from studies using ligand-binding assays that found that some CSPs in other Lepidopterans have chemosensory roles including in *Mamestra brassicae* (*Jacquin-Joly et al., 2001*), *B. mori* (*Qiao et al., 2013*), and *S. inferens* (*Zhang et al., 2014*). Compared to the *AlOBP* genes highly expressed in the antennae, only three *AlCSP*s had antennae-biased expression, indicating that these three genes may be involved in the recognition and transmission of sex pheromones, host volatiles, and other odorants. On the other hand, many insect CSPs are broadly expressed in non-chemosensory tissues and have non-chemosensory functions,
such as SexiCSP3, which has been shown to have effects on the survival and reproduction of *S. exigua* (*Gong et al., 2012*), and AmelCSP5, which is involved in embryonic integument formation in *A. mellifera* (*Foret, Wanner & Maleszka, 2007*). In this study, many *AlCSPs* were found in various tissues and were highly expressed in non-chemosensory tissues suggesting that these AlCSPs (especially AlCSP14, which had the highest expression) may be involved in other physiological functions apart from chemosensory ones.

Furthermore, we found that there were eight *AlOBPs* (28.5% of all AlOBPs) and six *AlCSPs* (30.0% of all AlCSPs) that displayed proboscis-biased expression. OBP and CSP gene expression in the proboscis has been observed in other insects including *Apolygus lucorum* (*Hua et al., 2012*), *Lygus lineolaris* (*Hull, Perera & Snodgrass, 2014*), *S. podoptera* (*Liu et al., 2015c*), and *A. dissimilis* (*Sun et al., 2016*). Further functional studies have also confirmed the gustation function of some genes: OBP49a in *D. melanogaster* is involved in the suppression of sweet taste by bitter chemicals (*Jeong et al., 2013*); some OBPs in *D. melanogaster* can modulate sucrose intake in response to a panel of nine bitter compounds by RNAi-mediated methods (*Swarup et al., 2014*); and CSP4, which is exclusively presented in the proboscis of two sibling species—*H. armigera* and *H. assulta*—an act as a wetting agent to reduce the surface tension of aqueous solutions (*Liu et al., 2014*). Therefore, the 14 AlOBPs and AlCSPs with proboscis-biased expression may play similar gustation functions in *A. lepigone*.

In the future, these AlOBPs and AlCSPs can help us develop environmentally friendly pesticides against *A. lepigone* based on reverse chemical ecology (*Dominguez et al., 2016*; *Zhou, 2010*). We can explore the functions of candidate OBPs/CSPs *in vitro* to screen compounds with high binding affinities (e.g., host plant volatiles or sex pheromones) to target the OBPs/CSPs. These compounds could then be investigated as potential pesticides or sexual attractants. Further, with genetic modification by the CRISPR/Cas9 editing system (*Hsu, Lander & Zhang, 2014*; *Li et al., 2016*; *Zhu et al., 2016*), we can knock out the candidate OBPs and CSPs to construct various mutant strains and then release the effective strains into the field to disrupt the chemical communication behaviors of the pest.

## CONCLUSION

In conclusion, we identified an extensive set of putative OBP- and CSP-encoding genes in *A. lepigone* based on our previous antennal transcriptomic data. As the first step towards understanding the functions of these genes, we conducted comprehensive and comparative phylogenetic analyses and developed gene expression profiles for OBPs and CSPs and found that most of the AlOBPs and AlCSPs had high identities with other moth odorant genes. Nearly half of the *AlOBP*s displayed antennae-biased expression, but many *AlCSP*s were detected in all tissues tested and were highly expressed in non-antennal tissues. Understanding the tissue and sex-biased expression patterns will help identify the functions of AlOBPs and AlCSPs, which in turn will aid in elucidating the chemosensory mechanism of *A. lepigone* and developing environmentally friendly pesticides against this pest in future.

## ACKNOWLEDGEMENTS

We thank Master students Mei-Yan Zheng (Nanjing Agricultural University, China), Bachelor students Cai-Yun Yin and Meng-Ya Li (Huaibei Normal University, China) for their help in collecting insects, and Dr. Peng He (Guizhou University, China) for suggestions to improve the manuscript.

### Funding

This work was supported by National Natural Science Foundation of China (31501647), Natura Science Fund of Education Department of Anhui province, China (KJ2017A387 and KJ2017B019), the Special Fund for Agro-scientific Research in the Public Interest of China (201303026), and Open Fund of Education Ministry Key Laboratory of Integrated Management of Crop Diseases and Pests of Nanjing Agricultural University. The funders had no role in study design, data collection and analysis, decision to publish, or preparation of the manuscript.

### Grant Disclosures

The following grant information was disclosed by the authors:
National Natural Science Foundation of China: 31501647.
Natura Science Fund of Education Department of Anhui province, China: KJ2017A387, KJ2017B019.
Agro-scientific Research in the Public Interest of China: 201303026.
Education Ministry Key Laboratory of Integrated Management of Crop Diseases and Pests of Nanjing Agricultural University.

### Competing Interests

The authors declare there are no competing interests.

### Author Contributions

- Ya-Nan Zhang conceived and designed the experiments, performed the experiments, analyzed the data, contributed reagents/materials/analysis tools, wrote the paper, prepared figures and/or tables, reviewed drafts of the paper.
- Xiu-Yun Zhu and Ji-Fang Ma analyzed the data, reviewed drafts of the paper.
- Zhi-Ping Dong and Long-Wa Zhang reviewed drafts of the paper.
- Ji-Wei Xu and Ke Kang performed the experiments, prepared figures and/or tables.

### DNA Deposition

The following information was supplied regarding the deposition of DNA sequences:
 All the gene sequences we identified in the paper are listed in Table S1.

### Data Availability

 The raw data has been supplied as a Supplementary File.

## Supplemental Information

Supplemental information for this article can be found online at http://dx.doi.org/10.7717/peerj.3157#supplemental-information.

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
