# Peer review of "Molecular identification and expression patterns of odorant binding protein and chemosensory protein genes in Athetis lepigone (Lepidoptera: Noctuidae)"

_PeerJ, doi:10.7717/peerj.3157_

## Round 0.1 · original submission · Major Revisions

The manuscript presents an interesting body of data, yet both reviewers agree that there are several points that should be addressed before the manuscript is suitable for publication; e.g, the discussion section where both reviewers suggest that it should be rewritten, paying special attention to the references mentioned by reviewer #2.

Reviewer 1 ·

Basic reporting

The manuscript “Molecular identification and expression patterns of odorant binding protein and chemosensory protein genes in Athetis lepigone (Lepidoptera: Noctuidae) “by Zhang YN et al describes the identification of 28 OBPs and 20 CSPs from the transcriptome of Atheltis lepigone. Identified OBP and CSP genes were further studied by qRT-PCR to compare their expression levels in male and female antennae and other tissues. This manuscript is suitable for publication in Peer J, but there are some issues that need to be addressed before it is acceptable for publication.

1. The generated transcriptome data will be of great interest to other researchers, I suggest the authors should submit the whole sequencing data to a public database. There is no reference to any submission in such database, even in the authors’ previous publication.

2. Discussion part is nearly a repeat of the Results part. The authors need to rewrite the Discussion part or combine Results and Discussions.

Experimental design

3. The authors claimed “this study help developing environment-friendly pesticides against this pest in future”. But How? How the identification of OBP/CSP from Atheltis lepigone will lead to environmentally friendly insect control strategies? Need more discussion.

Validity of the findings

4. Line 222, “17 AlOBPs (including PBPs and GOBPs) were highly expressed in the antennae”. How do you decide it is “highly” or “not highly”?

5.In discussion, L241 “The differences in gene number may be due to the different chemosensory behavior of different moths requiring distinct molecular mechanisms that have developed over evolutionary time.” Actually more B. mori OBP and CSP genes were identified because of the available genome sequences. Compare to only transcriptome results, you have chance to identify more genes from genome database.

Additional comments

6. From the transcriptome data, could authors do an in silico expression analysis to compare the expression levels of OBPs and CSPs between male and female antennae? It will be interesting to compare the expression profile obtained from transcriptome to the qPCR results?
7. Line 213 “AlCSP1/3/6/8/9/1012/15/17” need one / between 10 and 12.

·

Basic reporting

1. My main and most important comment is that the discussion should be significantly revised and improved. I have broken down suggestions on improvement of the discussion into the following sub points:

a. The motif pattern analysis is not mentioned in the discussion and its significance is not described. I suggest explicitly stating that the motif analysis is of the amino acid sequences, because that is not clearly stated, and motif analysis can also suggest analysis of gene regulatory elements. It’s not clear how this experiment adds to this body of work. There should be some discussion of the meaning of these data in the discussion. One suggestion is removing the data, and including it in a later manuscript where the data are explored more. Alternatively, I would suggest conducting bioinformatic analysis on the patterns. For example, is there a set of motifs associated with male or female biased OBP genes? With leg versus wing versus antenna expression?

b. There are key references missing in the discussion, on the role of OBP76a, also known as LUSH, in Drosophila melanogaster. This is the best characterized OBP gene, and mutant alleles have been examined. There are several referencesd to these studies which ought to be included.

c. There is a recent, comprehensive analysis of OBP gene expression and function in the antenna of Drosophila melanogster, which ought to be included in the introduction/discussion:

Larter NK, Sun JS, Carlson JR. Organization and function of Drosophila odorant binding proteins. Elife. 2016 Nov 15;5. pii: e20242. doi: 10.7554/eLife.20242. PubMed PMID: 27845621


d. In lines 253-257, the authors conclude that because OBP genes were found to be expressed in wings and legs, they may have non-chemosensory function. However, there are several problems with this conclusion. First, lepitopdteran legs and wings most likely contain chemosensory sensilla, which are probably gustatory. Second, OPBs have been demonstrated to play a role in gustatory function; the reference for that is:

Jeong YT, Shim J, Oh SR, Yoon HI, Kim CH, Moon SJ, Montell C. 2013. An odorant-binding protein required for suppression of sweet taste by bitter chemicals. Neuron 79:725–737. doi: 10.1016/j.neuron.2013.06.025, PMID: 23972598

If an OBP gene truly has a non-chemosensory function, then the gene should be expressed in a tissue that lacks sensilla (such as testis and ovary). I suggest two further experiments in the experimental design section.

e. In lines 234-243, in the discussion, the language should be edited comparing gene numbers of the various species.

f. In lines 244-246 of the discussion, I would suggest the word may to the following sentences. The reason for this is that the exact roles and functions of OBPs are still unknown. “Many studies show that the insect OBPs were mainly expressed in the antennae of both sexes, and may play key roles in the process of host location, mating, and oviposition by accurately recognizing environmental odorants.”

2. In line 92, in the introduction, the authors state that “the ALCSPs had a ubiquitous expression characteristic,” This is unclear statement, and it must be phrased more precisely.


3. Format correction: for figure 5, the graph of OPB19 is missing the enclosing line.

4. In line 221, in the results, there seems to be an error. The authors state “the results showed that all the OBPs and CSPs were expressed in the adult antennae of A. lepigone” This is not consistent with expression patterns in Figures 5 and 6, where it appears that expression of some genes is very low/absent in antennae.

5. The following sentence in lines 265-266 should be rewritten for more clarity: “indicating that these genes may participate in the reorganization of sex-related odorants, and need further functional analyses in future.”

6. Both the abstract and conclusion mention that ALCIPs had ubiquitous expression patterns; this must be clarified and rewritten.

Experimental design

Overall, the experiments appear well-designed, executed, and analyzed. I have two main comments:

1. My main suggestion for aligning the experimental design with the conclusions is to add at least two more tissue types to those analyzed in Figures 5 and 6. I would suggest adding proboscis/mouthparts and ovary/testis. Due to previous observations that OBPs are expressed in gustatory tissue, and due to inclusion of wings and legs, it’s essential to also know patterns in the proboscis; this will give a more comlete view of possible expression in gustatory tissue. The inclusion of ovary/testis would be a tissue that has no chemosensory sensilla, and would allow you to conclude that OBPs may have non-chemosensory functions.

2. For figures 5 and 6, the axis scales are extremely different for each analyzed gene. Since these have been normalized to housekeeping genes, they should be relatively comparable. The scales on the graphs should be similar also. I suggest describing these differences in the results section, and discussing their implications. For example, GOBP1 appears relatively highly expressed, whereas OBP19 and OBP23 are at relatively low levels.

Validity of the findings

The main comments I have in this section are interspersed in the previous two sections. To reiterate, I suggest substantially rewriting the discussion, I suggest adding new tissues to the qRT-PCR experiment. I also suggest that the analysis of amino acid motifs be explained and possibly analyzed in more depth to justify why that experiment fits into this body of work.

Additional comments

This is a strong data set, that is a companion set to the publication, Zhang et al., 2016, Molecular identification and sex distribution of two chemosensory receptor families in A. lepigone by antennal transcriptome analysis. This work adds more insight into a Notuid moth, that has an interesting natural history and is of economic interest. My main comment is that the discussion should be substantially rewritten, language errors corrected, key citations added, and possibly the addition of more tissues tested for qRT-PCR.

---

## Round 0.2 · Minor Revisions

Please make the final two changes requested by the reviewer so that the revised paper can be accepted.

·

Basic reporting

Much improved over the previous submission. I only have two suggestions to add: line 93 needs a citation. I also suggest rephrasing the first paragraph of the discussion, lines 324-338; the comparison of gene number and species is confusing.

Otherwise, the discussion is much improved.

Experimental design

Good; I appreciate the addition of more tissues to the qPCR analysis.

Validity of the findings

I think the findings are valid.

Additional comments

The manuscript is much improved and ready for publication. I have only two very minor suggestions, as stated elsewhere in the review, but printed here again for clarity: line 93 needs a citation. I also suggest rephrasing the first paragraph of the discussion, lines 324-338; the comparison of gene number and species is confusing.

---

## Round 0.3 · accepted · Accept

Congratulations! You paper is now acceptable for publication in PeerJ.